# Melatonin Modulates the Microenvironment of Glioblastoma Multiforme by Targeting Sirtuin 1

**DOI:** 10.3390/nu11061343

**Published:** 2019-06-14

**Authors:** Sheng-Wei Lai, Yu-Shu Liu, Dah-Yuu Lu, Cheng-Fang Tsai

**Affiliations:** 1Graduate Institute of Basic Medical Science, China Medical University, Taichung 40402, Taiwan; wayson081024@gmail.com; 2Department of Pharmacology, School of Medicine, China Medical University, Taichung 40402, Taiwan; yushuliu220@gmail.com; 3Department of Photonics and Communication Engineering, Asia University, Taichung 41354, Taiwan; 4Department of Biotechnology, Asia University, Taichung 41354, Taiwan

**Keywords:** SIRT1, glioblastoma, tumor microenvironment, infiltrating monocytes

## Abstract

Natural products have historically been regarded as an important resource of therapeutic agents. Resveratrol and melatonin have been shown to increase SIRT1 activity and stimulate deacetylation. Glioblastoma multiforme (GBM) is the deadliest of malignant types of tumor in the central nervous system (CNS) and their biological features make treatment difficult. In the glioma microenvironment, infiltrating immune cells has been shown to possess beneficial effects for tumor progression. We analyzed SIRT1, CCL2, VCAM-1 and ICAM-1 in human glioma cell lines by immunoblotting. The correlation between those markers and clinico-pathological grade of glioma patients were assessed by the Gene Expression Omnibus (GEO) datasets analysis. We also used monocyte-binding assay to study the effects of melatonin on monocyte adhesion to GBM. Importantly, overexpression of SIRT1 by genetic modification or treatment of melatonin significantly downregulated the adhesion molecular VCAM-1 and ICAM-1 expression in GBM. CCL2-mediated monocyte adhesion and expression of VCAM-1 and ICAM-1 were regulated through SIRT1 signaling. SIRT1 is an important modulator of monocytes interaction with GBM that gives the possibility of improved therapies for GBM. Hence, this study provides a novel treatment strategy for the understanding of microenvironment changes in tumor progression.

## 1. Introduction

SIRT1 also known as sirtuin 1, is one of the sirtuin family that belongs to class III histone deacetylases (HDACs) and show different functions and structure [1]. In mammals, there are seven members in the sirtuin family (SIRT1–SIRT7) which show different functions and structure [1,2]. SIRT1 is a NAD^+^-dependent deacetylase that removes an acetyl group from various histone and non-histone proteins [3]. SIRT1 is mostly well-known in its roles in aging and longevity, and is characterized as a chromatin-silencing component that represses gene transcription [4,5]. SIRT1-mediated deacetylation suppresses the function of several transcription factors, such as FOXO1, p53, STAT3 and NF-kB [6,7]. Recent cancer studies have shown that SIRT1 is a reliable biomarker of cancer recurrence and implied some anticancer properties of SIRT1 agonist or resveratrol [8,9]. In the tumor microenvironment, SIRT1 was engaged in the immune response through the activation of pro-inflammatory pathways [10]. Accumulating evidence has shown that sirtuin overexpression inhibits cell growth, induces apoptosis in glioma cell lines, including U87 and T98G [11]. Importantly, the level of SIRT1 expression is gradually decreased with advanced pathology grade of glioma patients and is significantly lower than normal brain tissue [12,13].

Melatonin (N-acetyl-5-methoxytryptamine) is synthesized from tryptophan and is mainly produced by the pineal gland [14,15]. It has been classically associated with circadian regulation [16]. One important characteristic of melatonin is that it readily passes through the blood-brain-barrier (BBB) and then accumulates in the central nervous system (CNS) [17]. Previous studies reported that melatonin possesses potent anti-inflammatory and anti-oxidant capability [18]. Several studies have showed that melatonin reverses acute and chronic inflammation, and significantly reduces some pro-inflammatory cytokines such as tumor necrosis factor (TNF)-α and interleukin (IL)-8 [19,20]. It has been reported that the neuroprotective effects of melatonin against acute neuroinflammation is via the SIRT1 signaling pathway [21]. In recent years, increasing evidences have shown that melatonin exerts an inhibitory effect on many cancer types such as gastric, colon and breast cancers [22,23]. Moreover, melatonin has also demonstrated that it can inhibit glioma cell invasion and migration through modulating oxidative stress pathways [24,25]. Clinical studies have shown an improvement in glioma patient survival rate by using radiotherapy plus melatonin compared with radiotherapy alone [26]. Currently, there have been several studies devoted to the possible interaction between melatonin and the immune system [27,28]. A previous study showed that melatonin enhances both natural and acquired immunity in animals [29]. Melatonin directly binds to melatonin receptors on T helper cells and modulates the immune response [30]. It also activates natural killer cells to enhance immune responses [31].

Glioblastoma multiforme (GBM) is one of the most malignant types of CNS tumors because their biological features lead to difficult therapeutic treatment [32,33]. Combination therapy, consisting of surgical resection followed by combined radiotherapy and chemotherapy represents the standard for patients with diagnosed glioblastoma [34]. Despite research advances, most GBM patients die within two years after diagnosis, resulting in a median survival of about 15 months [35]. GBM exhibits highly intratumoral heterogeneity, leading to resistance and eventual tumor recurrence [36]. In addition, abundant populations of infiltrating monocytes/macrophages constitute up to 40% of tumor mass and provide protection from immune surveillance. [37]. In the glioma microenvironment, infiltrating immune cells have demonstrated to possess beneficial effects for tumor progression [38,39]. Importantly, it has been reported that glioma-associated microglia may secrete immunosuppressive factors that do not participate in immune responses for anticancer effects [40]. Moreover, recent studies have shown that the crosstalk between infiltration immune cells and tumor cells is mediated by various soluble factors [41]. However, when circulating monocytes reach the tumor site, they are surrounded by several molecules and differentiate into cells needed by the tumor [42]. A recent study reported that reducing the effects of the monocyte of tumor-promotion in GBM can be a potential strategy for therapies [43]. Thus, understanding the characteristics of monocytes infiltration to glioma may identify therapeutic strategies that can be combined with standard therapies.

C-C motif ligand 2 (CCL2) chemokine also called monocyte chemoattractant protein-1 (MCP-1). CCR2 and CCR4 are two cell surface receptors that bind to CCL2. Under inflammatory state, CCL2 play critical roles in the activation and recruitment of inflammatory and immune cells [44,45]. During inflammation, resident cells including astrocytes, microglia, endothelial cells, infiltrating lymphocytes and macrophages constitutively express chemokine CCL2 in CNS [46,47]. Recently, it has been shown that the CCL2 expression level correlates with the recruitment of monocytes/macrophages into tumor tissues [48]. In addition, several studies have shown that an overexpression of CCL2 is correlated with metastasis and tumorigenesis in a variety of cancers, including breast cancer and glioblastoma [49,50]. Notably, CCL2 together with its cognate receptor CCR2 have shown to play important roles in cancer cell survival [51]. It has also been reported that CCL2 bind to CCR4 on lymphocytes, resulting in their recruitment to melanoma cells [52]. In the tumor microenvironment, the expression of CCL2 respond to pro-inflammatory cytokines or macrophage infiltration [53,54]. Overexpression of CCL2 was found in glioma cells and human tissues [55,56]. It has also been reported that pathologic grades of glioma are significantly related with expression levels of CCL2 [57]. In addition, tumor presenting high level of CCL2 have significantly poor outcome in GBM patients [57]. GBM cells secreted CCL2 that increase cancer invasion, migration and cell growth in an autocrine manner [57]. The above studies showed that CCL2 might play a critical role in tumor progression.

This study indicated that melatonin reduced the ICAM-1, VCAM-1 as well as chemokine CCL2 expression in GBM. Melatonin attenuates the monocyte binding activity. Importantly, we found that administration of melatonin increased the expression of SIRT1. We observed that treatment of SIRT1 activator or an overexpression of SIRT1 significantly reduced adhesion molecular ICAM-1 and VCAM-1 expression. Therefore, CCL2 involved in the expression of ICAM-1 and VCAM-1 promotes monocyte adhesion to GBM. In this study, we suggested that SIRT1 plays a critical role between monocytes and GBM interaction. Based on our results, it might help identify novel targets and improved therapies for GBM.

## 2. Materials and Methods

### 2.1. Materials

Melatonin, SB203580 (p38 inhibitor), SP600125 (JNK inhibitor) and BAY 11-7082 (NF-kB inhibitor) were obtained from Sigma-Aldrich. IL-1β was obtained from PeproTech. TPCK (IKB-α proteolysis inhibitor) and U0126 (MEK1/2 inhibitor) were obtained from Calbiochem. C021 (CCR4 antagonist) and RS102895 (CCR2 antagonist) were obtained from R&D Systems. CAY10591 (SIRT1 activator) was obtained from Cayman Chemicals. EX527 (inhibitor activator) was obtained from Tocris Bioscience. Primary antibodies against for β-actin and VCAM-1 were obtained from Abcam. Primary antibodies against for p38, ICAM-1, p65, PKCδ, ERK2, SIRT1, JNK1/3, p-PKCδ and p-ERK were obtained from Santa Cruz Biotechnology. Primary antibodies against for p-p65, p-p38 and p-JNK were obtained from cell signaling technology. Primary antibodies against for α-tubulin were obtained from Sigma-Aldrich. Neutralizing antibodies against human CCL2/MCP-1 (MAB279) were obtained from R&D Systems. On-target smart pool siRNA against ICAM-1 and VCAM-1 or non-targeting control siRNA were obtained from Dharmacon.

### 2.2. Cell Culture

Human glioma cells (U251) were purchased from the JCRB. Human glioma cells (U87), human glioma cells (A172), mouse glioma cells (ALTS1C1) and human monocytes (THP-1) were purchased from the BCRC. SIRT1-overexpression cells were generated in a laboratory of ours. Briefly, the puro plasmids of pCruz-SIRT1 were transfected into mouse glioma cells ALTS1C1. After two weeks, we had a selection of stable clones by using 500 μg/mL of neomycin, which were then maintained in neomycin (500 μg/mL) in growth medium. The SIRT1-overexpression cells were seeded in a 100-mm dish and were left growing for a few weeks. ALTS1C1 multiple clones were generated to stably express SIRT1, then the SIRT1 expression was checked by immunoblotting. The U251 and U87 GBM cell lines were grown in minimum essential medium (MEM), The A172 and ALTS1C1 glioma cell lines were grown in Dulbecco’s modified eagle medium (DMEM), and human monocyte cell line THP-1 was grown in RPMI-1640 medium. All the culture cells were grown in medium containing 10% fetal bovine serum (FBS), 100 mg/mL streptomycin and 100 U/mL penicillin (PS). All the cells were incubated at 37 °C in a humidified atmosphere containing 5% CO_2_ and 95% air.

### 2.3. SRB Assay

After treatment with melatonin, we aspirated the culture medium and followed by fixing it with 10% trichloroacetic acid (TCA) in GBM for 10 min. Then, 0.4% of Sulforhodamine B (SRB) was dissolved in 1% acetic acid and added to 96-well for 1 h. After the cells were stained for 40 min, unbound dye was washed twice by 1% acetic acid and the bound-SRB cells were resolved by Tris solution (10 mM). The absorbance of the resultant solution was read using a microplate reader and measured OD values at 515 nm.

### 2.4. MTT Assay

The protocol of 3-(4,5-dimethylthiazol-2-yl)- 2,5-diphenyltetrazolium bromide (MTT) assay was performed according to previous study of ours [58]. Briefly, GBM cells were cultured in 96-well plates and a growth medium was aspirated after indicated treatment. Cell were incubated with serum free medium containing MTT solution (0.5 µg/mL) at 37 °C for 1 h. After removing the medium, we added 150 µL of DMSO into each 96-well and shook it for 20 min. Occasionally, pipetting of the liquid resolved the MTT formazan. The absorbance was measured with a microplate reader and measured OD values at 515 nm.

### 2.5. Cytosolic and Nuclear Extracts

Nuclear extracts were prepared as previously described [59]. Briefly, cells were rinsed with cold PBS and resuspended in a hypotonic buffer (10 mM HEPES, pH 7.6, 10 mM KCl, 1 mM dithiothreitol, 0.1 mM EDTA, and protease inhibitor cocktail) for 10 min on ice. The cytosolic proteins were separated using centrifugation at 10,000× *g* for 2 min. The supernatants containing the cytosolic proteins were collected, and the pellets containing the nuclear fraction were resuspended in buffer (20 mM HEPES pH 7.6, 1 mM EDTA, 1 mM dithiothreitol, 0.4 M NaCl, 25% glycerol, and protease inhibitor cocktail) for 30 min on ice. The suspensions were centrifuged again at 13,000× *g* for 20 min, and the supernatants containing the nuclear proteins were collected and stored at −80 °C.

### 2.6. Monocyte-Binding Assay

Human monocyte THP-1 cells were incubated with 0.1 μg fluorescent dye of BCECF/AM (2’,7’-bis-(2-carboxyethyl)-5-(and-6)-carboxyfluorescein) in a RPMI-1640 medium in the incubator for 1 h. GBM cells were administrated with IL-1β or melatonin for the different time periods. Then, the medium was removed from the 6-wells, the monolayer of GBM cells were added with 2.0 × 10^5^ BCECF/AM-labeled-THP-1 cells to each 6-well. We removed the non-adherent monocytes and gently washed twice them with culture medium. After 45 min incubated. The adherent monocytes were then photographed and calculated using a fluorescence microscope.

### 2.7. Western Blotting

Whole cell extracts were performed in accordance to previous study. Briefly, GBM cells were extracted with cell lysis buffer (RIPA) and using a scraper to collect the cells, which were then kept on ice. The protein samples were spun at 12,000 × rpm for 30 min. We collected the supernatant and then stored it at −20 °C. We then separated the 30 μg of protein samples by running SDS-page, then transferred them onto PVDF membranes. Afterwards, we blocked membranes with non-fat dry milk (5%) in TBST for 1 h. The membrane was incubated with primary antibodies at 4 °C overnight or RT for 1 h. Following washes with TBST buffer, the membranes were incubated with anti-mouse or anti-rabbit HRP-conjugates secondary antibodies. Protein bands were visualized by ECL and Kodak X-OMAT LS film. The data was quantified using an ImageJ software.

### 2.8. Reverse Transcription and Real-Time PCR

Total RNA was isolated from GBM cells using TRIzol (TRI Reagent) and the concentration of RNA was measured with the BioDrop spectrophotometer. The interest gene expression was detected by quantitative real-time PCR (q-PCR). The messenger RNA was converted into cDNA by a reverse transcription (RT) reaction process using the invitrogen reverse transcription kit and amplified using the oligonucleotide primers as following: CCL2; ICAM-1; VCAM-1 and internal control 36B4. PCR reaction using SYBR Green qPCR Master Mix was performed in experiments on the StepOne Plus Real-Time PCR Systems.

### 2.9. Cell Transfection

The GBM cells were transiently transfected with 10nM siRNA (Dharmacon) against ICAM-1 and VCAM-1. Then, control siRNA was carried out using Lipofectamine 3000 at 37 °C 24 h. Lipofectamine 3000 and target siRNA were premixed in serum-free medium for 10 min before being used for cell transfection. After 24 h incubation, the medium containing Lipofectamine 3000 was replaced with fresh serum-free medium.

### 2.10. Reporter Gene Assay

The GBM cells were transiently transfected with Renilla luciferase plasmid (0.1 μg) and CCL2 promoter luciferase plasmid (1 μg). Added with reporter lysis buffer into each 6-well, and the protein samples were collected by spin at 12,000× rpm for 20 min. Luciferase activity was measured by a dual-luciferase reporter assay system and the values were normalized by a Renilla luciferase.

### 2.11. Enzyme-Linked Immunosorbent Assay (ELISA)

Mini ELISA development kits were used to detect human CCL2 expression by GBM cells. Buffers used throughout this protocol were purchased as an ELISA Buffer Kit from R&D Systems (Catalog #DY279-05). GBM cells were grown in a serum-free medium with or without IL-1β or melatonin. After 24 h, we collected the medium (100 microliter in 96-well) for ELISA assay according to the manufacturer’s instructions.

### 2.12. GEO Gene Expression Database

The DNA microarray data were sourced from the datasets of glioma patients. The expression levels of the target gene were analyzed using GraphPad Prism 6 software from the publicly available Gene Expression Omnibus (GEO) databases. The glioma patients were collected from the Henry Ford Hospital (HFH) which contained 180 glioma patients with histologically confirmed different grades of glioma: grade four astrocytomas (GBM *n* = 81), grade three (astrocytomas *n* = 19, oligodendrogliomas *n* = 12), grade two (astrocytomas *n* = 7, oligodendrogliomas *n* = 38) and non-tumors *n* = 23. The gene expression of SIRT1, ICAM-1 and VCAM-1 values were obtained from the GSE4290 dataset and we evaluated the correlation with human glioma pathological grade.

### 2.13. Statistics

The results present the mean ± S.E.M. and all the data were performed with at least three biologically independent replicates. The values were determined using ImageJ software, SigmaPlot software (version 10.0, Systat Software Inc., San Jose, CA, USA) and GraphPad Prism 6 software (version 6, GraphPad software Inc., San Diego, CA, USA). The data given are statistical analysis between two samples that were performed using a Student’s *t*-test. One-way ANOVA followed by the Bonferonni multiple comparison test was used where indicated. In all cases, a *p*-value < 0.05 was considered to be of statistical significance. The *p*-values are indicated in the figure legends. No pre-test was used to choose sample size. No data points were excluded.

## 3. Results

### 3.1. IL-1β Induces VCAM-1 and ICAM-1 Expression and Increases Monocyte Adhesion in GBM

In our previous study, we showed that TNF-α induces expression of VCAM-1 on the GBM surface. These results provide evidence that monocyte through the VCAM-1 adhere to GBM. Moreover, the VCAM-1 levels positive correlated with the glioma pathological grade [60]. First, we analyze the human glioma microarray datasets. GSE4290 indicated that VCAM-1 and ICAM-1 levels were higher in the grade four glioma group than in the low-grade glioma and non-tumor group (Figure 1A,B). We further determined the effect of cytokines on adhesion molecules ICAM-1 and VCAM-1 expression in GBM. The results showed that cytokines IL-1β and TNF-α strongly increased ICAM-1 and VCAM-1 expression (Figure 1C). Importantly, IL-1β is a major mediator of inflammatory cytokine, which is produced by activated macrophages [61].

We further examined the effects of IL-1β on ICAM-1 and VCAM-1 expression in GBM. The expression of ICAM-1 and VCAM-1 were markedly induced by IL-1β in different GBM cells (Figure 2A). We further determined the monocytes binding activity in GBM by using monocyte-binding assay. Treated with IL-1β increased THP-1 monocyte adhesion to GBM (green color; Figure 2B and Appendix A). The IL-1β-induced ICAM-1 and VCAM-1 mRNA expression were also observed in GBMs (Figure 2C). We further investigated the relationship between ICAM-1 and VCAM-1 levels and numbers of adherent human monocytes in GBMs. Transfection with siRNA against VCAM-1 or ICAM-1 significantly decreased the expression of ICAM-1 and VCAM-1 (Figure 2D). In addition, the increasing ability of monocytes adhesion to GBM was decreased by transfection with ICAM-1 or VCAM-1 siRNA (Figure 2E and Appendix A). These results showed that monocyte adhesion to GBM occurs via ICAM-1 and VCAM-1.

### 3.2. Effects of Melatonin in IL-1β-Induced ICAM-1 and VCAM-1 Expression and Monocyte Adhesion

We further assessed the regulatory effects of melatonin on GBM. The concentrations of melatonin used in the present study were according to previous reports [62,63]. In this study, we also tested the viability of a wide range of concentrations (0, 0.25, 0.5, 1 or 3 mM) of melatonin. Melatonin treatment (ranging from 0 to 3 mM) did not affect the cell viability when compared with the control group (Appendix A). In addition, melatonin dramatically inhibited IL-1β-induced ICAM-1 and VCAM-1 expression in a dose-dependent manner (Figure 3A). Furthermore, treatment of melatonin significantly attenuated the IL-1β-increased monocyte binding activity in a dose-dependent manner (Figure 3B and Appendix A). Similarly, melatonin markedly downregulated the mRNA levels of ICAM-1 and VCAM-1 induced by IL-1β (Figure 3C). These results revealed that melatonin effectively reduces the enhancement of ICAM-1 and VCAM-1 expression in GBM.

### 3.3. Upregulation of SIRT1 Inhibits ICAM-1 and VCAM-1 Expression in GBM

It has been reported that the upregulation of SIRT1 by melatonin was able to increase the deacetylation of several SIRT1 substrates [64]. Therefore, the SIRT1 inhibitor impairs the beneficial action of melatonin on cell viability prevention. First, we analyzed the human glioma microarray GSE4290 dataset which demonstrated that the levels of SIRT1 were lower in the GBM group than in the low-grade glioma and non-tumor group (Figure 4A). We further investigated whether the induction of SIRT1 by melatonin mediates the downregulation of ICAM-1 and VCAM-1 expression in GBM. U251 and U87 treated with melatonin increased SIRT1 protein levels in a dose-dependent (Figure 4B) and time-dependent manner (Figure 4C). Importantly, administration of CAY10591 (SIRT1 activator) effectively antagonized the IL-1β-induced ICAM-1 and VCAM-1 protein expression (Figure 4D). In contrast, treatment with EX527 (SIRT1 inhibitor) resulted in higher expressions of ICAM-1 and VCAM-1 induced by IL-1β (Figure 4E). However, both CAY10591 and EX527 did not affect SIRT1 expression. This means that SIRT1 is involved in regulation of IL-1β-induced ICAM-1 and VCAM-1 expression. Importantly, transfection with wild-type SIRT1 significantly decreased the IL-1β-induced VCAM-1 expression. (Figure 4F). The above results indicated that SIRT1 is a critical modulator of ICAM-1 and VCAM-1 expression in GBM.

### 3.4. Involvement of CCL2 in the IL-1β-Induced ICAM-1 and VCAM-1 Expression and Monocyte Adhesion

We have observed the effects of IL-1β on CCL2 expression in GBM. IL-1β induced high CCL2 expression in U251 and U87 (Figure 5A). Furthermore, the IL-1β-induced CCL2 protein secretion was found in both U251 and U87 in a dose-dependent manner (Figure 5B). 

CCL2 has been reported to bind its receptors CCR2 and CCR4 on lymphocytes, resulting in recruiting immune cells to tumor sites, which causes immunosuppression [65]. We next determined whether CCL2 is involved in ICAM-1 and VCAM-1 expression in GBM. Treated with CCR2 antagonist RS102895 or CCR4 antagonist C 021 effectively reduced human monocyte adhesion (Figure 6A). Moreover, CCR2 and CCR4 antagonist administration also reduced IL-1β-enhanced ICAM-1 and VCAM-1 levels in GBM (Figure 6B,C). Similar effects of the inhibitors were observed using neutralizing antibodies, where the anti-CCL2 neutralizing antibodies attenuates the IL-1β-induced ICAM-1 and VCAM-1 expression (Figure 6D). These results showed that CCL2 is a critical modulator of monocyte adhesion to GBM as well as ICAM-1 and VCAM-1 expression.

### 3.5. Involvement of p38/p65 in the IL-1β-Induced VCAM-1 and ICAM-1 in GBM

IL-1β-stimulation increased phosphorylation of p65 and MAP kinase in a time-dependent manner (Figure 7A,C). There has been reported that TPCK inhibited expression of inflammatory mediators by NF-kB activation and directly blocked IKK activity [66,67]. Moreover, administration of NF-kB related pharmacological inhibitor, TPCK (IKB-α proteolysis inhibitor) or BAY11-7082 (NF-kB inhibitor), effectively attenuates the IL-1β-enhanced VCAM-1 and ICAM-1 protein levels (Figure 7B). Moreover, treatment with the MAP kinase pharmacological inhibitor, SB203580 (p38 inhibitor), SP600125 (JNK inhibitor) and U0126 (MEK1/2 inhibitor) antagonized the IL-1β-induced VCAM-1 and ICAM-1 expression (Figure 7D). Notably, administration with SB203580 effectively inhibited IL-1β-induced VCAM-1 and ICAM-1 expression. Interestingly, treatment with various pharmacological inhibitors did not affect SIRT1 expression in GBM (Figure 7B,D). Additionally, treatment with IL-1β resulted in an accumulation of NF-kB subunits p50/p65 in the nucleus. However, NF-kB translocation into the nucleus was inhibited significantly by melatonin (Figure 7E). These results indicated that the IL-1β-induced VCAM-1 and ICAM-1 in GBM is mediated through the p38/p65 pathways.

## 4. Discussion and Conclusions

Research has been published showing that chemokine CCL2 plays a critical role in GBM progression [68]. A previous study showed that CCL2 binding to cell surface CCR4 or CCR2 receptor that affects cell migration [69]. Another report suggested that the CCL2-mediated expression of ICAM-1 on the human lymphatic endothelial cells was effectively inhibited by the CCL2 neutralizing antibody [70]. Furthermore, CCL2 also increases VCAM-1 expression in human fibroblasts [71]. Glioma cells have shown to secrete various soluble factors, including CCL2, which contributes to immune surveillance [72]. Importantly, overexpression of CCL2 was found in GBM human tissues [73]. This reinforces the findings from other groups that CCL2 induction is correlated to the glioma with poor outcomes [57]. In this study, we found that IL-1β-induced ICAM-1 and VCAM-1 expression and monocyte adhesion, which is modulated by CCL2. Furthermore, IL-1β-induced CCL2 expression mediated monocyte adhesion through CCR2 and CCR4 axis.

Another study showed that melatonin administration increases the survival rate of glioma patients combined with radiotherapy [74]. Furthermore, researchers found that melatonin suppresses self-renewal and tumorigenic activity of glioma stem cells (GSCs). A recent study showed that treatment with melatonin can interrupt the interaction of endothelial cells and neutrophils mediated by ICAM-1 [75]. In addition, melatonin also modulates neuronal plasticity by regulating neural cell adhesion molecules expression in the brain [76]. A previous study reported that melatonin could balance the lipopolysaccharide (LPS)-induced CCL2 expression, and these effects are accompanied with the anti-inflammatory cytokine IL-10 production [77]. Our study supports that melatonin inhibits IL-1β-induced ICAM-1 and VCAM-1 expression. We also suggested that melatonin downregulated IL-1β-induced CCL2 transcriptional activity in GBM cells. It has been shown that melatonin is able to increase the several substrates of SIRT1 deacetylation [64].

Increasing evidence suggests a role for NF-kB and MAP kinase pathways in the pathogenesis of GBM and its resistance to treatment, indicating that it may be useful targets for treatment [78,79]. Moreover, MAP Kinase pathway is correlated with gliomagenesis and is associated with poorer prognosis of glioma patients [80]. Recently, our reports have shown that NF-kB signaling pathway play a key role of glioma cell motility in response to growth factors [81]. In addition, our previous study suggested that NF-kB inhibitor (PDTC), and IkB protease inhibitor (TPCK) inhibited the potentiating action of TNF-α [82]. It has been shown that the treatment of U251 cells with MAP kinase p38-siRNA inhibited proliferation and induced apoptosis [83]. Currently, our study also showed that the VCAM-1-associated monocyte adhesion to GBM is mediated through the MAP kinase p38 pathway [84]. Our study supports that cytokine IL-1β-stimulation increased phosphorylation of p65 and MAP kinase and promoted NF-kB translocation into the nucleus.

Accumulation reports supporting a role for SIRT1 in the GBM progression [85]. Recent studies showed that overexpression of sirtuin in glioma cells could inhibit growth and proliferation, as well as increase apoptosis [11]. It has been shown that SIRT2 is downregulated in melanomas and gastric carcinomas [86]. In addition, SIRT6 is downregulated in colon adenocarcinoma and pancreatic cancer [87]. Our study indicated that the expression level of SIRT1 was gradually downregulated with advanced pathology grade of glioma patients, and was significantly lower than normal brain tissues. Current studies are examining the biological functions of SIRT1 activators with the aim of identifying cancer treatments [88]. In the present study, we found that an upregulation of SIRT1 in melatonin treated GBM. Clinically, melatonin has been reported to inhibit glioma growth in combination with chemotherapeutics and radiation therapy [26,89]. Recent studies showed that SIRT1 suppressed the adhesion molecules expression by suppressing inflammatory signaling [90,91]. Interestingly, expression levels of ICAM-1 and VCAM-1 was increased in SIRT1 +/− compared to SIRT1 wild type mice in atherosclerotic plaques. These findings showed that SIRT1 prevents adhesion molecule expression [91]. Our study supports that SIRT1 activators or overexpression of SIRT1 effectively inhibited the adhesion molecules in GBM.

In conclusion, our results indicate that adhesion molecular VCAM-1 and ICAM-1 are critical modulators of the CCL2-dependent monocytes interaction with GBM. Based on the understanding of pathologic mechanisms of GBM from our previous studies, monocytes interacting with GBM increases the M1 pro-inflammatory cytokines level. When monocytes bind to the GBM, it further enhances cell proliferation and promotes adhesion molecules expression on GBM, then increases more monocytes adhesion. This study also indicated that IL-1β resulted in an accumulation of NF-kB subunits p50/p65 in the nucleus, which further enhances CCL2 expression. These CCL2-dependent phenomena were mediated through the p38/p65 pathways. GBM secretes CCL2 and further modulates VCAM-1 and ICAM-1 expressions via CCR2 and CCR4 axis. The adherent monocyte then secrets more IL-1β to stimulate cancer cells to form a positive feedback loop (Figure 8). Hence, in this study, we provide a novel treatment strategy for understanding the microenvironment changes in tumor progression.

## Figures and Tables

**Figure 1 nutrients-11-01343-f001:**
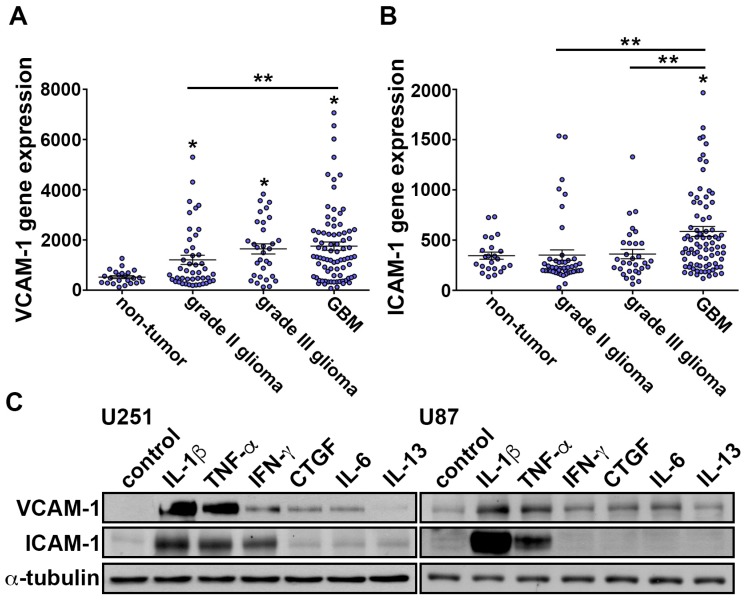
VCAM-1 and ICAM-1 levels correlated with the glioma clinico-pathological grade. (**A**) Messenger RNA levels of VCAM-1 in specimens of glioma patients that were obtained from GSE4290 datasets. The data are presented as mean ± S.E.M * *p* < 0.05 compared with the non-tumor group. ** *p* < 0.05 grade II glioma group compared with the GBM group. (**B**) Messenger RNA levels of ICAM-1 in specimens from glioma patients that were obtained from GSE4290 datasets. The values are presented as mean ± S.E.M. * *p* < 0.05 compared with the non-tumor group. ** *p* < 0.05 grade II and grade III glioma group compared with the GBM group. (**C**) U251 and U87 cells were added with IL-1β (10 ng/mL), TNF-α (10 ng/mL), IFN-γ (50 ng/mL), CTGF (100 ng/mL), IL-6 (10 ng/mL) or IL-13 (10 ng/mL) cytokines for 24 h. The expressions levels of ICAM-1 and VCAM-1 were analyzed using western blotting.

**Figure 2 nutrients-11-01343-f002:**
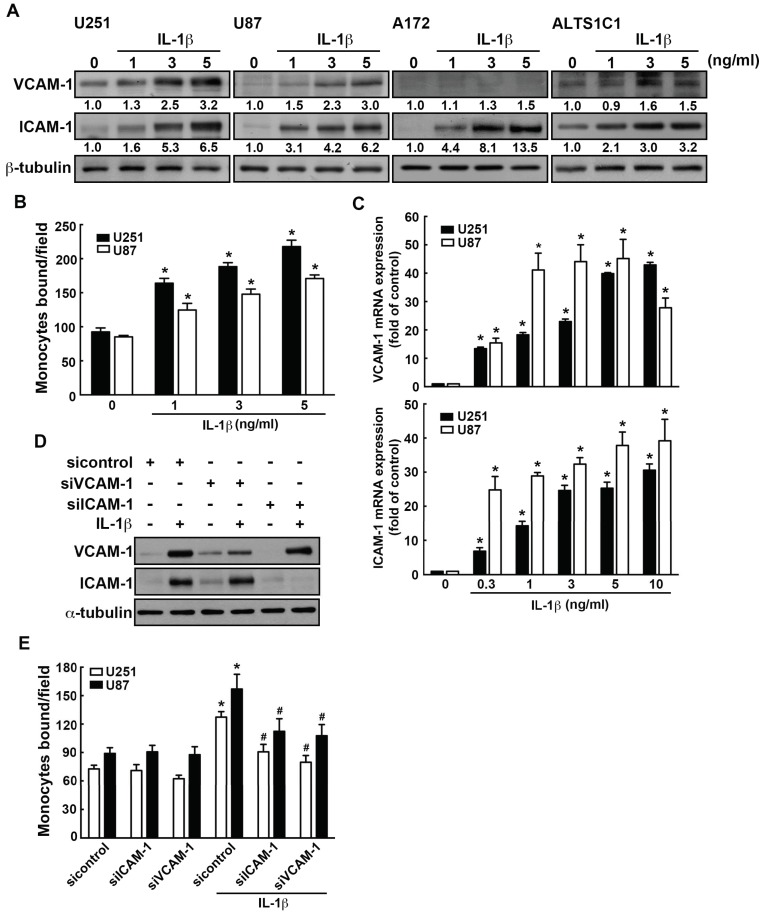
IL-1β induces VCAM-1 and ICAM-1 expression and increases monocyte adhesion in GBM. (**A**) Four different GBM cells U251, U87, A172 or ALTS1C1 were treated with various concentration of IL-1β (1, 3, or 5 ng/mL) for 24 h. The expression levels of ICAM-1 and VCAM-1 were assessed by western blotting. (**B**) GBM cells U251 and U87 were added with various concentrations of IL-1β (1, 3, or 5 ng/mL) for 24 h. BCECF-AM- labeled-THP-1 were added to GBM cells for 45 min, and then the adherence of THP-1 was analyzed using fluorescence microscopy. One-way ANOVA followed by Bonferonni multiple comparison test was used to determine the significance of the data. * *p* < 0.05 compared with the control group. The data are presented as mean ± S.E.M. (representative of independent experiments = 3) (**C**) Cells were treated with various concentrations of IL-1β (0.3, 1, 3, 5, or 10 ng/mL) for 6 h, and VCAM-1 and ICAM-1 expression was assessed by real-time PCR. One-way ANOVA followed by Bonferonni multiple comparison test was used to determine the significance of the data. * *p* < 0.05 compared with the control group. The data are presented as mean ± S.E.M. (representative of independent experiments = 3). (**D**) U251 GBM cells were transfected with siRNA against control, ICAM-1, or VCAM-1 for 24 h and added with IL-1β (3 ng/mL) for another 24 h. VCAM-1 and ICAM-1 expression, and monocyte adhesion ability were assessed by monocyte-binding assay (**E**) The values are presented as mean ± S.E.M (representative of independent experiments = 3). * *p* < 0.05 compared with the control siRNA group. # *p* < 0.05 compared with the IL-1β administration group.

**Figure 3 nutrients-11-01343-f003:**
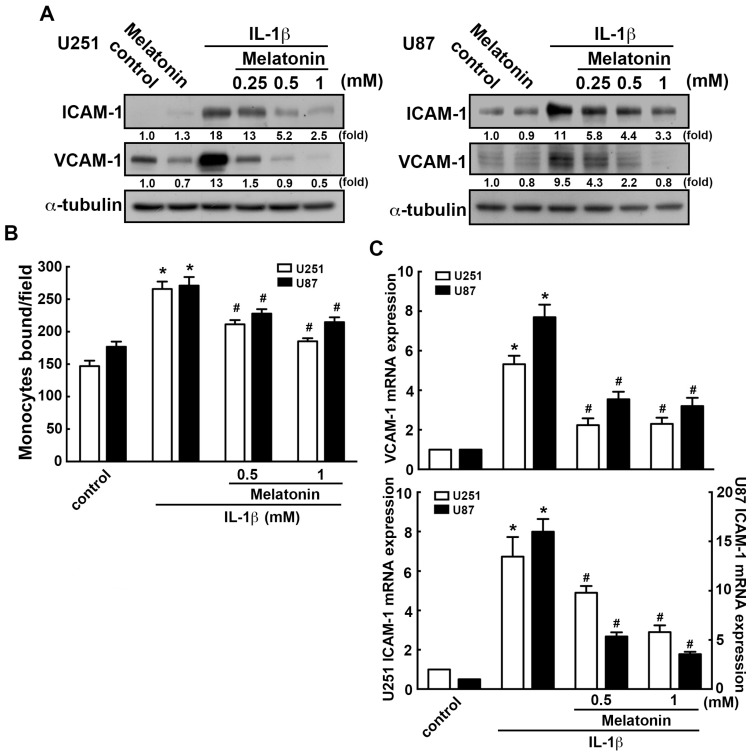
Melatonin attenuates IL-1β-induced ICAM-1 and VCAM-1 expression in GBM. U251 and U87 cells were pretreated with various concentration of melatonin (0.25, 0.5 or 1 mM) for 45 min, then added to IL-1β (3 ng/mL) for another 24 h (**A**), or 6 h (**C**). ICAM-1 and VCAM-1 expression were analyzed by western blotting (**A**) and real-time PCR (**C**), respectively. (**B**) U251 and U87 cells were pretreated with melatonin (0.5 or 1 mM) for 45 min then added with IL-1β (3 ng/mL) for another 24 h. The binding activity of monocyte were analyzed by evaluating the BCECF-AM-labeled-THP-1 monocytes by the fluorescence microscopy. The data are presented as mean ± S.E.M. (representative of independent experiments = 3). * *p* < 0.05 compared with the control group. # *p* < 0.05 compared with the IL-1β treatment group.

**Figure 4 nutrients-11-01343-f004:**
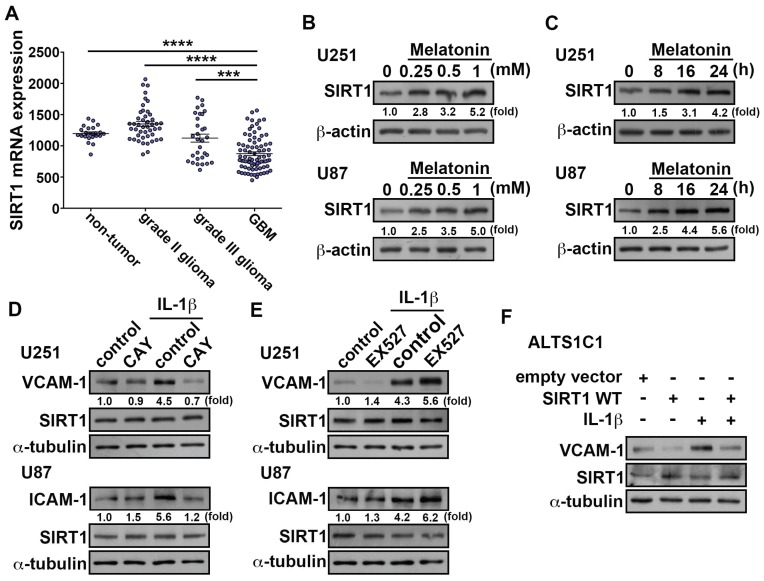
Upregulation of SIRT1 attenuates IL-1β-induced ICAM-1 and VCAM-1 expression in GBM. (**A**) Messenger RNA levels of SIRT1 in glioma patient specimens obtained from GSE4290 datasets. The data are presented as mean ± S.E.M. **** *p* < 0.0001 grade II glioma group or non-tumor group compared with GBM group. *** *p* < 0.05 grade III glioma group compared with the GBM group. (**B**) GBM cells U251 and U87 were administrated with various concentrations of melatonin (0.25, 0.5 or 1 mM) for 24 h, and expression of SIRT1 was analyzed using western blot. (**C**) GBM cells U251 and U87 cells were treated with melatonin (0.5 mM) for different time periods (8, 16 or 24 h), and SIRT1 expression was analyzed using western blotting. Cells were pretreated with SIRT1 activator CAY10591 (5 μM) (**D**) or SIRT1 inhibitor EX527 (10 μM) (**E**) for 45 min then administrated with IL-1β (3 ng/mL) for another 24 h. The expression levels of SIRT1, ICAM-1 and VCAM-1 and expression was assessed using western blotting. (**F**) ALTS1C1 were transfected with wild-type SIRT1 or empty vector for 24 h and administrated with IL-1β (3 ng/mL) for another 24 h. The expression of VCAM-1 and SIRT1 were assessed using western blotting. The data are presented as mean ± S.E.M. (representative of independent experiments = 3). WT: wild type.

**Figure 5 nutrients-11-01343-f005:**
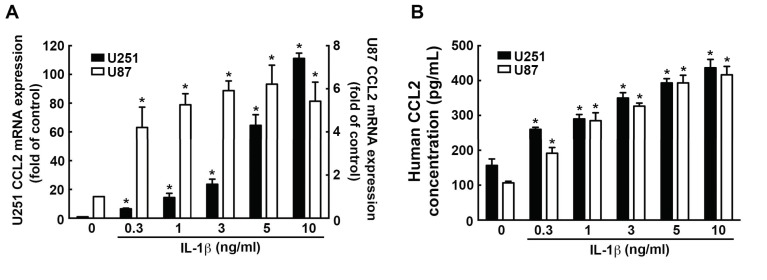
Upregulation of CCL2 expression induced by IL-1β in GBM. Cells were treated with various concentrations of IL-1β (0.3, 1, 3, 5, or 10 ng/mL) for 6 h (**A**) or 24 h (**B**), and CCL2 expression was analyzed by real-time PCR (**A**) and ELISA (**B**). One-way ANOVA followed by Bonferonni multiple comparison test was used to determine the significance of the data. * *p* < 0.05 compared with the control group. The data are presented as mean ± S.E.M. (representative of independent experiments = 3).

**Figure 6 nutrients-11-01343-f006:**
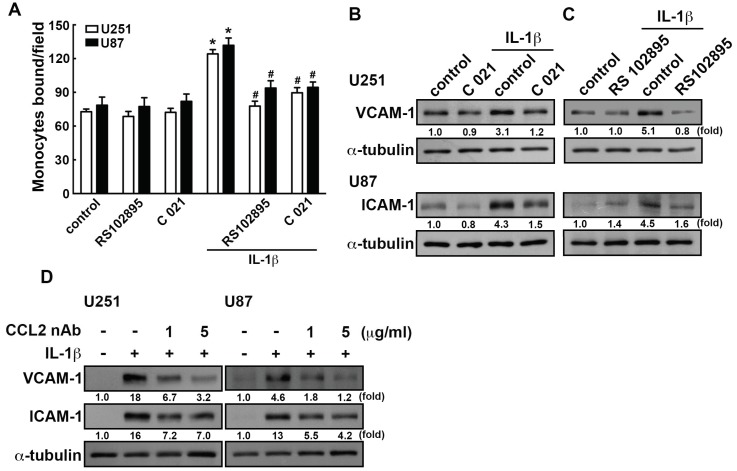
CCL2 involves in the IL-1β-induced ICAM-1 and VCAM-1 in GBM. (**A**) U251 and U87 cells were pretreated with CCR2 antagonist RS102895 (400 nM) or CCR4 antagonist C 021 (400 nM) for 45 min and administrated with IL-1β (3 ng/mL) for another 24 h. The binding activity of monocyte were analyzed by evaluating the BCECF-AM-labeled-THP-1 using the fluorescence microscopy. GBM cells were added with CCR4 antagonist C 021 (400 nM) (**B**) or CCR2 antagonist RS102895 (400 nM) (**C**) for 45 min and added with IL-1β (3 ng/mL) for another 24 h. ICAM-1 and VCAM-1 expression was assessed by western blotting. (**D**) U251 and U87 cells were administrated with anti-CCL2 neutralizing antibodies (1 or 5 μg/mL) for 45 min and added with IL-1β (3 ng/mL) for another 24 h. The expression levels of ICAM-1 and VCAM-1 was assessed by western blotting. The values are presented as mean ± S.E.M. (representative of independent experiments = 3). * *p* < 0.05 compared with the control group, # *p* < 0.05 compared with the IL-1β administration group. nAb, neutralizing antibody.

**Figure 7 nutrients-11-01343-f007:**
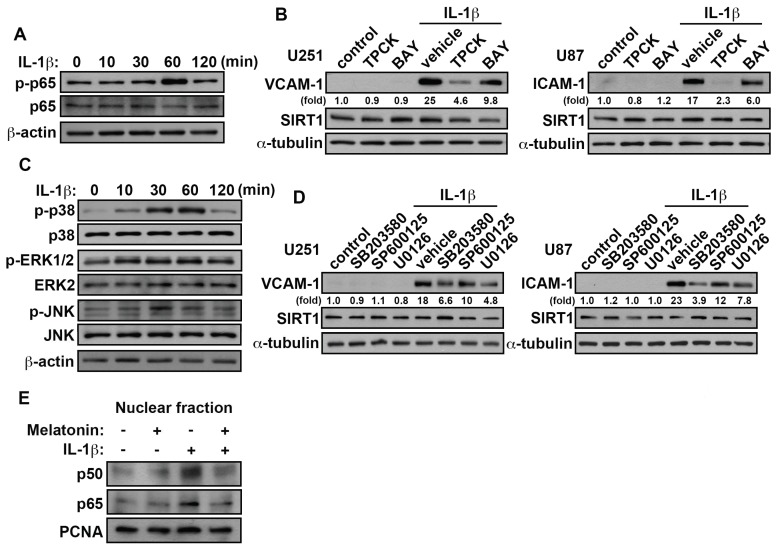
p38/p65 signaling pathways are involved in the VCAM-1 and ICAM-1 expression were induced by IL-1β in GBM. (**A**) GBM cells were added with IL-1β (3 ng/mL) for different time periods (10, 30, 60 or 120 min) and p-p65 expression were analyzed by western blotting. (**B**) Cells were added to TPCK (10 μM) and BAY 11-7082 (3 μM) for 45 min then administrated with IL-1β (3 ng/mL) for another 24 h. The expression levels of VCAM-1, ICAM-1 and SIRT1 were analyzed by western blotting. (**C**) GBM cells were added with IL-1β (3 ng/mL) for various time periods (10, 30, 60 or 120 min), and p-p38, p-ERK1/2 and p-JNK expression was assessed using western blotting. (**D**) U251 and U87 were added with SB203550 (10 μM), SP600125 (10 μM) and U0126 (1 μM) for 45 min and added with IL-1β (3 ng/mL) for another 24 h. The expression levels of SIRT1, VCAM-1 and ICAM-1 were assessed using western blotting. (**E**) GBM cells were pretreated with melatonin (0.5 mM) for 45 min and added with IL-1β (3 ng/mL) for another 2 h. The nuclear extracts from GBM cells were subjected to western blotting. Expression levels of p50 and p65 was determined using western blotting. PCNA was used as nuclear internal controls. The values are presented as mean ± S.E.M. (representative of independent experiments = 3).

**Figure 8 nutrients-11-01343-f008:**
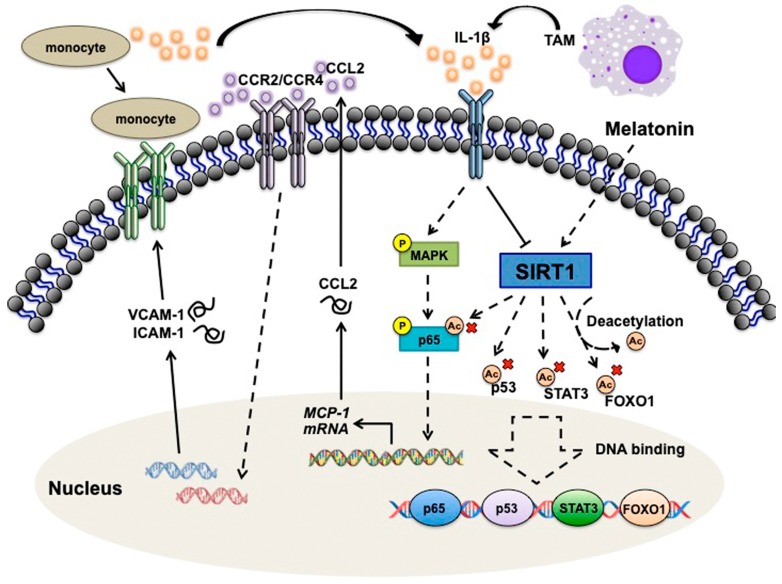
Schematic diagrams of the signaling pathways which are involved in the SIRT1 modulates microenvironment of GBM. IL-1β releases from tumor-associated macrophage, then activates the p38 and p65 signaling. This results in an accumulation of NF-kB in the nucleus which further enhances CCL2 expression. GBM secretes CCL2 and further modulates VCAM-1 and ICAM-1 expressions via CCR2 and CCR4 axis. The VCAM-1 and ICAM-1 expression on GBM surface then subsequently enhances monocyte binding to GBM. The adherent monocyte then secretes more IL-1β to stimulate cancer cells to form a positive feedback loop. Melatonin could induce SIRT1 upregulation and attenuate cytokines and adhesion molecules expression.

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
