# Peer review of "Melatonin Modulates the Microenvironment of Glioblastoma Multiforme by Targeting Sirtuin 1"

_nutrients, 2019, doi:10.3390/nu11061343_

Reviewer 1 Report

Lai et al. analyzed the involvement of microenvironment changes in tumor progression. This manuscript was well-written and well-designed. However, there was some serious weakness regarding statics and its physiological role.

Major points

1) The authors performed Student t-test for statics. However, t-test was not suitable statics for multiple comparisons. The authors should perform adequate statics, such as Dunnet, Bonferonni, or Tukey throughout the manuscript.

2) The authors analyzed the molecular aspects of several cytokines and melatonin. However, its physiological or functional roles were not analyzed. If the authors want to discuss the microenvironment changes and tumor progression, the authors should add some functional analysis which relates to tumor progression. Because the authors only perform in vitro analysis, this reviewer recommends performing in vivo analysis.

3) In Fig.2 knockdown efficiencies of siICAM-1 and siVCAM-1 were required.

Minor points

4) In Fig. 3, the concentration of melatonin was too high. The authors should discuss this point.

Author Response

Reply to Reviewer #1:

Reviewer #1

Major points:

Point 1. The authors performed Student t-test for statics. However, t-test was not suitable statics for multiple comparisons. The authors should perform adequate statics, such as Dunnet, Bonferonni, or Tukey throughout the manuscript.

Response 1: We appreciate the reviewer’s suggestion. Following to the advice, we have re-analyzed the quantitative results and statistical analysis by Bonferonni correction. One-way ANOVA assessed statistical analysis between three or more independent group. In all cases, a value of p less than 0.05 was considered significant.

 Point 2. The authors analyzed the molecular aspects of several cytokines and melatonin. However, its physiological or functional roles were not analyzed. If the authors want to discuss the microenvironment changes and tumor progression, the authors should add some functional analysis which relates to tumor progression. Because the authors only perform in vitro analysis, this reviewer recommends performing in vivo analysis.

Response 2: We really appreciate the reviewer’s suggestion. In our recent study, we used three-dimensional monocyte-GBM co-culture system for looking the regulatory effects between GBM cells and monocytes. The results have showed that monocyte adhesion to GBM that promotes tumor growth and invasion activity (Liu YS et al., Oncogene, 2017).

In addition, the adherent monocytes also dramatically increased the levels of M1 markers TNF-α and IFN-γ, but not those of M2 markers ARG1, IL-10 and CD163 in the GBM-monocyte co-culture model. Our results suggest that monocyte adhesion to GBM might increase adhesion molecule expression in GBM with subsequent monocyte adhesion in a positive feedback loop, which leads to tumor progression (Liu YS et al., Oncogene, 2017). Based on our previous results, this study further investigated the regulatory effects of SIRT1 and identify the effects of chemokine CCL2 in recruited monocyte in GBM.

 Point 3. In Fig.2 knockdown efficiencies of siICAM-1 and siVCAM-1 were required.

Response 3: We appreciate the reviewer’s comment. According to the advice, we have added new experiments to confirm the knockdown efficiencies of siICAM-1 and siVCAM-1in Figure 2D. (page 7)

 Minor points:

Point 4. In Fig. 3, the concentration of melatonin was too high. The authors should discuss this point.

Response 4: We are grateful to the reviewer’s comment. The concentrations of melatonin used in the present study were according to previous reports (XIAOTING WANG et al., 2018; JIAGUI QU et al., 2013). In this study, we also tested the viability of a wide range of concentrations (0, 0.25, 0.5, 1 or 3 mM) of melatonin. We also performed the SRB and MTT experiments using melatonin with various concentrations. Melatonin treatment (ranging from 0 to 3 mM) did not affect the cell viability when comparing with control group (Supplementary Fgure 2A).

References:

1. Wang, X.; Wang, B.; Xie, J.; Hou, D.; Zhang, H.; Huang, H. Melatonin inhibits epithelialtomesenchymal transition in gastric cancer cells via attenuation of IL1beta/NFkappaB/MMP2/MMP9 signaling. Int J Mol Med 2018, 42, 2221-2228, doi:10.3892/ijmm.2018.3788.

2. Qu, J.; Rizak, J.D.; Li, X.; Li, J.; Ma, Y. Melatonin treatment increases the transcription of cell proliferation-related genes prior to inducing cell death in C6 glioma cells in vitro. Oncol Lett 2013, 6, 347-352, doi:10.3892/ol.2013.1413.

Reviewer 2 Report

The aim of this manuscript was to elucidate the role of melatonin in SIRT1 modulation, which led to microenviromental changes in GBM and caused inhibition of tumour progression by modulation of monocytes interaction with GBM. It has been earlier shown that the level of SIRT1 decreased with advanced stage of glioma patients in comparison to normal brain tissue. Moreover, the Authors showed that VCAM1 and ICAM1 are activated by NFkB. Rationales for the research were justified. The introduction describes the current knowledge in the studied area and supports the undertaken topic. The presented research seems to be important and justified. The manuscript is mostly prepared carefully, written quite well and the content is interesting and clearly present. The results are convincing and interesting, presented in a logical manner, confirmed on several GMB cell lines. The interpretation of data is mostly adequate (the exceptions are described below).

Comments:

The photos presented in figures 2B, 2D, 3D and 6A are unreadable, too small, and it is difficult to interpret them. In my opinion, the graphs are enough to show the results.

Line 275 – the description of SIRT1 level is a little confusing. Please, describe it in a more precise manner.

Description of figure 4C – there is no information, which concentration of melatonin was used (the time period is given twice).

Figure 4F – confusing data repetition or not precise description (maybe more time points were analyzed).

Figure 4 and 6BD –It will be better to change the description “vehicle” to “control”.

The interpretation of figure 4D is not convincing, because the anti-CCL2 neutralizing antibodies did not reduce the level of ICAM1. I can agree that the effect was observed for VCAM1 but not for ICAM1, where the level is almost the same. It is an overinterpretation and this issue should be described more precisely.

The specificity of MAP kinases and NF-kB inhibitors should be described. Especially in the case of NFkB inhibitor, TPCK, the activity of which is not limited to this transcription factor (e.g. TNF is also its target)

The interpretation concerning the role of MAP kinases and NFkB in VCAM1 and ICAM1 activation should be a little bit toned. The usage of an inhibitor of MAP, only slightly decreased the level of VCAM1 and ICAM1. The role of NFkB in this process is not enough evidenced, because the analysis of the phosphorylation of p65 only is not enough to tell about NFkB activation (it would be helpful to show, for example, nuclear or cytoplasmic localization) and inhibitors, as it was mentioned above, are not specific enough.

Rottlerin is mentioned in Material and methods. There was no information later on in the text about its use. Why was it mentioned?

Where were the inhibitors purchased or from whom were they received? No information about the suppliers/source is included in the text in the case of CAY, EX527, CO21, RS102895, TPCK, BAY. Moreover, short characteristics of all of them will be desirable.

In the paragraph Material and methods, there is a description of the MMT assay, but this method was not used in the description of the results.

Description for figure 8 is not precise and not clear enough. It should be extended/completed and all the dependencies between cell/secreted proteins should be considered.

Discussion – line 366 – “In addition, treatment with CCL2 upregulated VCAM1 and ICAM1…” Something is wrong.

Some editorial and linguistic corrections are required.

Author Response

Reply to Reviewer #2:

Reviewer #2 (Comments and Suggestions for Authors):

Point 1. The photos presented in figures 2B, 2D, 3D and 6A are unreadable, too small, and it is difficult to interpret them. In my opinion, the graphs are enough to show the results.

Response 1: We really appreciate the reviewer’s suggestion. Following to the advice, we have removed the photos in Figure 2B, 2D, 3D and 6A.

 Point 2. Line 275 – the description of SIRT1 level is a little confusing. Please, describe it in a more precise manner.

Response 2: We are grateful for the reviewer’s suggestion. We have revised the description in line 419 and 423.

Administration of CAY10591 (SIRT1 activator) effectively antagonized the IL-1β-induced ICAM-1 and VCAM-1 protein expression (Figure. 4D). In contrast, treatment with EX527 (SIRT1 inhibitor) resulted in higher expressions of ICAM-1 and VCAM-1 induced by IL-1β (Figure. 4E). However, both CAY10591 and EX527 did not affect SIRT1 expression. (page 9)

 Point 3. Description of figure 4C – there is no information, which concentration of melatonin was used (the time period is given twice).

Response 3: We really appreciate the reviewer’s suggestion. We have added the concentration of melatonin in legend of Figure 4C section. (page 9)

 Point 4. Figure 4F – confusing data repetition or not precise description (maybe more time points were analyzed).

Response 4: We are grateful to the reviewer for the comments. We have revised the presentation in Figure 4F to make it clearly. (page 9)

 Point 5. Figure 4 and 6BD –It will be better to change the description “vehicle” to “control”.

Response 5: We appreciate the reviewer’s suggestion. We have changed “vehicle” into “control” throughout the manuscript.

 Point 6. The interpretation of figure 4D is not convincing, because the anti-CCL2 neutralizing antibodies did not reduce the level of ICAM1. I can agree that the effect was observed for VCAM1 but not for ICAM1, where the level is almost the same. It is an overinterpretation and this issue should be described more precisely.

Response 6: The description of experiments of the anti-CCL2 neutralizing antibodies was written in lines 487-489 (pages 10).

We using the different concentration of anti-CCL2 neutralizing antibodies (1 or 5 μg/ml) attenuates the IL-1β-induced ICAM-1 and VCAM-1 expression (Figure. 6D). In U251 cell, IL-1β treatment group increased approximately 18-fold of VCAM-1 and 16-fold of ICAM-1 compared with control group. While cells exposure of anti-CCL2 neutralizing antibodies (1 μg/ml) decreased approximately 6.7-fold of VCAM-1 and 7.2-fold of ICAM-1 compared with IL-1β treatment group. Similarly, treatment with anti-CCL2 neutralizing antibodies (5 μg/ml) attenuated about 3.2-fold of VCAM-1 and 7.0-fold of ICAM-1 lower than IL-1β treatment group (Figure 6D, left panel). Similar effects of the neutralizing antibodies were observed in U87 cell (Figure 6D, right panel).

 Point 7. The specificity of MAP kinases and NF-kB inhibitors should be described. Especially in the case of NFkB inhibitor, TPCK, the activity of which is not limited to this transcription factor (e.g. TNF is also its target).

Response 7: We really appreciate the reviewer’s comment and we have revised and discussed the role of MAP kinases and NF-kB inhibitors in Discussion section. (page 14)

 Point 8. The interpretation concerning the role of MAP kinases and NFkB in VCAM1 and ICAM1 activation should be a little bit toned. The usage of an inhibitor of MAP, only slightly decreased the level of VCAM1 and ICAM1. The role of NFkB in this process is not enough evidenced, because the analysis of the phosphorylation of p65 only is not enough to tell about NFkB activation (it would be helpful to show, for example, nuclear or cytoplasmic localization) and inhibitors, as it was mentioned above, are not specific enough.

Response 8: We really appreciate the reviewer’s comment and we have stated the modulation of MAP kinase and NF-kB in VCAM-1 and ICAM-1 activation in Results (page 11) and Discussion sections (page 14).

According to the advice, we have added the experiments of cytosolic and nuclear extracts. The results showed that treatment with IL-1β resulted in an accumulation of NF-kB subunits p50/p65 in the nucleus. However, NF-kB translocation into the nucleus was inhibited significantly by melatonin (Figure. 7E). These results indicated that the IL-1β-induced VCAM-1 and ICAM-1 in GBM is mediated through NF-kB signaling. (lines 522-524, page 12)

 Point 9. Rottlerin is mentioned in Material and methods. There was no information later on in the text about its use. Why was it mentioned?

Response 9: We apologize for the mistake. We have corrected the information in Material and methods section. (page 3)

 Point 10. Where were the inhibitors purchased or from whom were they received? No information about the suppliers/source is included in the text in the case of CAY, EX527, CO21, RS102895, TPCK, BAY. Moreover, short characteristics of all of them will be desirable.

Response 10: We are grateful to the reviewer for the comments on our manuscript. Following your suggestions, we have added the information and revised the Materials section. (lines 109-113, page 3)

 Point 11. In the paragraph Material and methods, there is a description of the MMT assay, but this method was not used in the description of the results.

Response 11: We are grateful to the reviewer’s comments. In this study, we tested the viability of a wide range of concentrations (0, 0.25, 0.5, 1 or 3 mM) of melatonin. We also performed the SRB and MTT experiments using melatonin with various concentrations. Melatonin treatment (ranging from 0 to 3 mM) did not affect the cell viability when comparing with control group (Supplementary Figure 2A).

 Point 12. Description for figure 8 is not precise and not clear enough. It should be extended/completed and all the dependencies between cell/secreted proteins should be considered.

Response 12: We appreciate the reviewer’s suggestion. According to the advice, we have revised the description of Figure 8. (page 13)

We found that the IL-1β could release from tumor-associated macrophage then activates p38 and p65 signaling. Resulted in an accumulation of NF-kB in the nucleus which further enhance CCL2 expression. GBM secrets CCL2 and further modulates VCAM-1 and ICAM-1 expressions via CCR2 and CCR4 axis. The VCAM-1 and ICAM-1 expression on GBM surface then subsequently enhances monocyte binding to GBM. The adherent monocyte then secrets more IL-1β to stimulate cancer cells to form a positive feedback loop. Melatonin could induce SIRT1 upregulation and attenuate cytokines and adhesion molecules expression. (line 571-578, page 13)

 Point 13. Discussion – line 366 – “In addition, treatment with CCL2 upregulated VCAM1 and ICAM1…” Something is wrong.

Response 13: We apologize for the mistake and the sentence has been corrected. (page 13)

 Point 14. Some editorial and linguistic corrections are required.

Response 14: We really appreciate to the reviewer’s comments. We have corrected the grammatical errors and spellings throughout the text.

 Reference:

Liu, Y.S.; Lin, H.Y.; Lai, S.W.; Huang, C.Y.; Huang, B.R.; Chen, P.Y.; Wei, K.C.; Lu, D.Y. MiR-181b modulates EGFR-dependent VCAM-1 expression and monocyte adhesion in glioblastoma. Oncogene 2017, 36, 5006-5022, doi:10.1038/onc.2017.129.

Round  2

Reviewer 1 Report

The authors answered my questions.